# A Heteroscedastic Uncertainty Model for Decoupling Sources of MRI Image Quality

**Richard Shaw**[1,2]                      RICHARD.SHAW.17@UCL.AC.UK

[1] *Dept. Medical Physics & Biomedical Engineering, University College London, UK*

[2] *School of Biomedical Engineering & Imaging Sciences, King's College London, UK*

**Carole H. Sudre**[2,1,3]                   CAROLE.SUDRE@KCL.AC.UK

[3] *Dementia Research Centre, Institute of Neurology, University College London, UK*

**Sébastien Ourselin**[2]               SEBASTIEN.OURSELIN@KCL.AC.UK

**M. Jorge Cardoso**[2]                 M.JORGE.CARDOSO@KCL.AC.UK

## Abstract

Quality control (QC) of medical images is essential to ensure that downstream analyses such as segmentation can be performed successfully. Currently, QC is predominantly performed visually at significant time and operator cost. We aim to automate the process by formulating a probabilistic network that estimates uncertainty through a heteroscedastic noise model, hence providing a proxy measure of task-specific image quality that is learnt directly from the data. By augmenting the training data with different types of simulated k-space artefacts, we propose a novel cascading CNN architecture based on a student-teacher framework to decouple sources of uncertainty related to different k-space augmentations in an entirely self-supervised manner. This enables us to predict separate uncertainty quantities for the different types of data degradation. While the uncertainty measures reflect the presence and severity of image artefacts, the network also provides the segmentation predictions given the quality of the data. We show models trained with simulated artefacts provide informative measures of uncertainty on real-world images and we validate our uncertainty predictions on problematic images identified by human-raters.

## 1. Introduction

Quality control (QC) in magnetic resonance imaging (MRI) is the process of establishing whether a scan or dataset meets a required set of standards. QC typically relates to the acceptable level of image quality required for a particular task, which may be affected by acquisition noise, resolution, and/or image artefacts induced for instance by blood, motion, bias field, zipper or radio-frequency (RF) spikes. In MRI, a large variety of potential artefacts need to be identified so that problematic images can either be excluded or accounted for in further image processing and analysis. To date, the gold standard for identification of these issues remains labour-intensive visual inspection of the data (Graham et al., 2018).

However, with the current trend towards acquiring and exploiting very large imaging datasets, the time and resources required to perform visual QC have become prohibitive. Furthermore, as with other rating tasks, visual QC is subject to inter and intra-rater variability due to differences in radiological training, rater competence, and sample appearance (Sudre et al., 2019).

Some artefacts, such as those caused by motion, can also be difficult to detect with visual QC, as their identification require the careful examination of every slice in a volume. These challenges have led to an increased interest in automated methods. In addition to the challenges inherent to visual QC, it is worth highlighting the task-dependent nature of a quality assessment: what may be deemed of acceptable quality for a radiological assessment may not be sufficient to provide reliable measurements for some of the automated analyses the image would undergo.

In this work, we propose to estimate task-specific uncertainty in a deep-learning framework and show this can be used as a measure of image quality for the task of segmentation. Furthermore, we show that we can decouple sources of uncertainty related to different imaging artefacts. Being able to decouple and identify sources of uncertainty can have a direct impact on the management of both clinical and research logistics. For instance, if the observed uncertainty is associated to acquisition artefacts (noise for instance) inspection of the scanner by an engineer may be required while if the uncertainty stems from subject motion, introducing ghosting or blurring, recall of the subject may be the appropriate path of action. In addition, in the context of population studies, producing the uncertainty associated with the desired measurement allows for appropriate statistical treatment of the samples while limiting the number of exclusions for quality reasons. Lastly, real-time indication of levels of uncertainty concerning a downstream task would enable radiographers to best manage session time and ensure repeat scans are taken as needed.

**Contributions** The main contributions of this work are three-fold:

1. A general method of estimating MR image quality in a self-supervised manner.

2. A novel cascading student-teacher CNN architecture and probabilistic loss function to decouple sources of uncertainty related to the task and different image artefacts.

3. Validation of uncertainty predictions on problematic images identified by expert raters.

## 2. Related Work

In recent years, estimating uncertainty in the data/model has become increasingly recognised as an important step to enable the safe transition of automated methods into the clinical environment (Wang et al., 2018), (Tanno et al., 2019). In Bayesian Deep Learning, two main types of uncertainty are commonly distinguished: *epistemic* uncertainty which is uncertainty in the model, and *aleatoric* uncertainty which depends on noise or randomness in the data. In a similar approach to the one presented in this work, Prado et al. (Prado et al., 2019) have used a dual network to learn both epistemic and aleatoric uncertainty, assuming their independence. However, since the focus of this work is on the assessment of image quality, only the aleatoric uncertainty is considered here.

## 3. A Heteroscedastic Aleatoric Uncertainty Model

Aleatoric uncertainty is classically divided into two categories; the *homoscedastic* component is the task-dependent uncertainty, while the *heteroscedastic* component depends on the input data, reflecting for instance its quality, and can be predicted as a model output. Following this classification, task-specific image quality is modelled according to a heteroscedastic

noise model. Heteroscedastic models assume that observation noise $\sigma^2$ can vary with the input $\mathbf{x}$, allowing for regions of the observation space to have higher noise levels than others (Kendall and Gal, 2017).

In this work, a single task, grey matter segmentation is considered; thus task uncertainty should be similar across experiments. The total predicted uncertainty is further assumed to be the sum of the task uncertainty (uncertainty given clean data) and the heteroscedastic uncertainties introduced as a function of image corruption.

For the segmentation task, the problem is presented as a voxel-wise classification. The likelihood to maximise is defined as the softmax function of the scaled output logits, i.e. $p(\mathbf{y}|\mathbf{f^W}(\mathbf{x}), \sigma) = \text{Softmax}\left(\mathbf{f^W}(\mathbf{x})/\sigma^2\right)$ where $\mathbf{f^W}(\mathbf{x})$ is the output of a neural network with weights $\mathbf{W}$ and input $\mathbf{x}$ (Bragman et al., 2018). The negative log likelihood is therefore:

$$-\log p(\mathbf{y} = c|\mathbf{f^W}(\mathbf{x}), \sigma) = -\log \text{Softmax}\left(\frac{1}{\sigma^2}\mathbf{f}_c^{\mathbf{W}}(\mathbf{x})\right) \tag{1}$$

$$= -\frac{1}{\sigma^2}\mathbf{f}_c^{\mathbf{W}}(\mathbf{x}) + \log \sum_{c'} \exp\left(\frac{1}{\sigma^2}\mathbf{f}_{c'}^{\mathbf{W}}(\mathbf{x})\right) \tag{2}$$

where $\mathbf{f}_c^{\mathbf{W}}(\mathbf{x})$ is the $c^{\text{th}}$ element of the output vector $\mathbf{f^W}(\mathbf{x})$. Note, in practice for segmentation we compute the unscaled cross entropy loss of $\mathbf{y}$, given by $\text{CE}\left(\mathbf{y} = c, \mathbf{f^W}(\mathbf{x})\right) = -\log \text{Softmax}\left(\mathbf{f}_c^{\mathbf{W}}(\mathbf{x})\right) = -\mathbf{f}_c^{\mathbf{W}}(\mathbf{x}) + \log \sum_{c'} \exp\left(\mathbf{f}_{c'}^{\mathbf{W}}(\mathbf{x})\right)$. Substituting this into Eq. 2:

$$-\log p(\mathbf{y} = c|\mathbf{f^W}(\mathbf{x}), \sigma) = \frac{1}{\sigma^2}\text{CE}\left(\mathbf{y} = c, \mathbf{f^W}(\mathbf{x})\right) + \log \frac{\sum_{c'} \exp\left(\frac{1}{\sigma^2}\mathbf{f}_{c'}^{\mathbf{W}}(\mathbf{x})\right)}{\left(\sum_{c'} \exp\left(\mathbf{f}_{c'}^{\mathbf{W}}(\mathbf{x})\right)\right)^{\frac{1}{\sigma^2}}} \tag{3}$$

Following (Kendall et al., 2017) the likelihood is approximated: $\left(\sum_{c'} \exp\left(\mathbf{f}_{c'}^{\mathbf{W}}(\mathbf{x})\right)\right)^{\frac{1}{\sigma^2}} \approx \frac{1}{\sigma}\sum_{c'} \exp\left(\frac{1}{\sigma^2}\mathbf{f}_{c'}^{\mathbf{W}}(\mathbf{x})\right)$. Substituting into Eq. 3 results in the weighted cross entropy loss that is used as the base loss function for all segmentation networks, as given by Eq. 4.

$$\mathcal{L}_{NN} = \frac{1}{\sigma^2}\text{CE}\left(\mathbf{y}, \mathbf{f^W}(\mathbf{x})\right) + \frac{1}{2}\log \sigma^2 \tag{4}$$

**Decoupling Multiple Uncertainties** We use the weighted cross entropy loss in Eq. 4 to learn voxel-wise uncertainty, i.e. the network has two outputs: the segmentation $\mathbf{y}$ and the variance $\sigma^2$, as shown by the task network in Figure 1. We adapt this loss function to predict multiple uncertainty quantities related to different aspects of image quality. The aim is to decompose the total predicted uncertainty $\sigma^2$ into multiple uncertainty quantities related to the inherent difficulty of the task and to different types of image degradation or augmentation that may affect image quality. Our model assumes the variance sum law for independent events, such that the total predicted variance is the sum of the individual variances associated with each mode of augmentation, i.e. $\sigma^2 = \sigma_t^2 + \sigma_1^2 + \sigma_2^2 + ... + \sigma_N^2 = \sigma_t^2 + \sum_{i=1}^{N}\sigma_i^2$ for $N$ possible augmentations, where $\sigma_t^2$ is the task uncertainty and $\sigma_i^2$ is the uncertainty due to the $i^{\text{th}}$ augmentation. This assumption of independence has the merit to simplify the model and ensure training tractability. While interactions with task uncertainty (task harder to learn with noisier data) or between degradation types (blurring

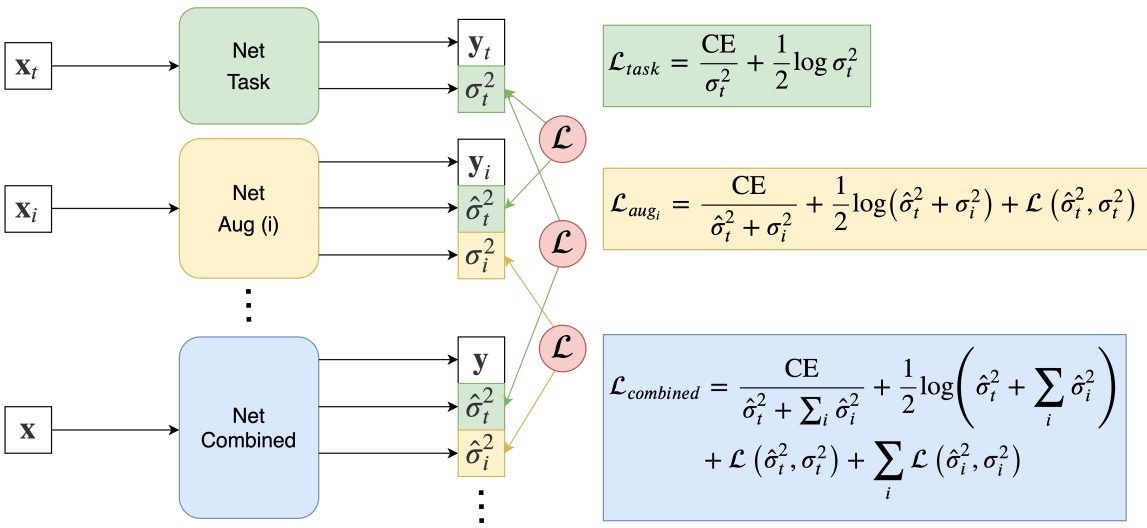

Figure 1: Proposed network architecture and training strategy: First, the task network is trained on clean images $\mathbf{x}_t$ to predict segmentation $\mathbf{y}_t$ and task uncertainty $\sigma_t^2$. Then, for each augmentation $i$, a teacher network is trained to predict both the task uncertainty $\hat{\sigma}_t^2$ and augmentation uncertainty $\sigma_i^2$, where the output from the task network supervises with consistency loss $\mathcal{L}(\hat{\sigma}_t^2, \sigma_t^2)$. Lastly, a combined student network is trained, where the uncertainty outputs from all previous teacher networks supervise its uncertainty predictions in a similar fashion. The loss functions for each CNN at each training stage are shown in corresponding colours.

and noise for instance) exist, their modelling would require the learning of new covariance terms and would greatly complexify both model and training procedure.

Substituting for $\sigma^2$ in Eq. 4 results in the combined loss function in Eq. 5.

$$\mathcal{L}_{combined} = \frac{\text{CE}\left(\mathbf{y}, \mathbf{f}^{\mathbf{W}}(\mathbf{x})\right)}{\sigma_t^2 + \sum_{i=1}^{N} \sigma_i^2} + \frac{1}{2} \log \left(\sigma_t^2 + \sum_{i=1}^{N} \sigma_i^2\right) \tag{5}$$

The purpose of $\mathcal{L}_{combined}$ is to enable the network to predict task uncertainty $\sigma_t^2$ and augmentation uncertainty $\sigma_i^2$ for each mode of augmentation. Since networks trained with this probabilistic loss function learn uncertainty in an unsupervised way, we do not have explicit labels for uncertainty. Therefore the network cannot determine how to decompose the total variance into separate quantities $\sigma_t^2$ and $\sigma_i^2$ by itself, i.e. without supervision. To do this, inspired by student-teacher networks (Tarvainen and Valpola, 2017) and knowledge distillation (Hinton et al., 2015), (Xie et al., 2019), we use a series of intermediate teacher networks where each network predicts the uncertainty due to a single augmentation, creating "pseudo labels" of uncertainty maps. By training these teacher networks sequentially, the output uncertainties from each intermediate network are used as self-supervising labels for the uncertainties predicted by a final combined student network. This is shown schematically in Figure 1.

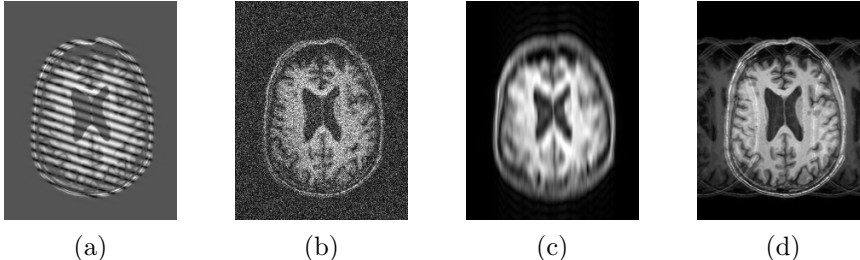

(a)      (b)      (c)      (d)

Figure 2: Examples of our k-space augmentations, a) RF spike artefact b) Gaussian noise in the k-space c) low-pass filter along one axis d) aliasing/wraparound artefact.

The training procedure can be summarised in the following three steps: 1) Train a teacher network on clean data only to predict the segmentation $\mathbf{y}_t$ and task uncertainty $\sigma_t^2$. 2) Freeze the task network and train a new network $\mathcal{N}_i$ for each mode of augmentation we wish to decouple, where each augmented network predicts the segmentation $\mathbf{y}_i$, the task uncertainty and the noise uncertainty $\sigma_i^2$ for augmentation $i$. The output uncertainty from the first network acts as a "pseudo label" for the task uncertainty. 3) Freeze all previous networks and train a final student network with all modes of data augmentation to predict the task uncertainty and all possible augmentation uncertainties, where each uncertainty is supervised by the pseudo uncertainty labels from their respective teacher networks.

For each network to learn the task uncertainty, an additional consistency loss term $\mathcal{L}(\hat{\sigma}_t^2, \sigma_t^2)$ is added to the weighted cross entropy loss to minimise the difference in uncertainty. Therefore, each augmentation network $\mathcal{N}_i$ minimises a loss function $\mathcal{L}_{aug_i}$ given by Eq. 6,

$$\mathcal{L}_{aug_i} = \frac{\text{CE}\left(\mathbf{y}_i, \mathbf{f}^{\mathbf{W}}(\mathbf{x})\right)}{\sigma_t^2 + \sigma_i^2} + \frac{1}{2}\log\left(\sigma_t^2 + \sigma_i^2\right) + \mathcal{L}(\hat{\sigma}_t^2, \sigma_t^2) \tag{6}$$

where,

$$\mathcal{L}(\hat{\sigma}^2, \sigma^2) = \mathcal{L}_1(\hat{\sigma}^2, \sigma^2) + \mathcal{L}_{grad}(\hat{\sigma}^2, \sigma^2) + \lambda\mathcal{L}_{SSIM}(\hat{\sigma}^2, \sigma^2). \tag{7}$$

$\mathcal{L}_1$ is the L1 loss of the uncertainty and $\mathcal{L}_{grad}$ is the L1 loss of gradient differences of the uncertainty maps in all three axes. The term $\mathcal{L}_{SSIM}$ computes the 3D structural similarity (SSIM). The gradient and structural similarity losses help preserve the structure of the predicted uncertainty maps as the level of degradation increases. However, in the presence of severe image artefacts, the position, shape, appearance/visibility of the segmentation boundary can change causing SSIM to breakdown. Therefore $\mathcal{L}_{SSIM}$ is down-weighted by $\lambda = 0.1$. A simplified SSIM with a $3 \times 3 \times 3$ average filter is used in our implementation.

## 4. k-Space Augmentation

During the training of each segmentation CNN, random k-space augmentations affecting image quality are applied on-the-fly. The types of augmentation are designed to emulate realistic MRI artefacts and are detailed below. Each augmentation is applied in the k-space by computing the 3D Fast Fourier Transform (FFT) of input volumes, modifying the k-space, and then computing the inverse 3D FFT, taking the magnitude image scaled

between 0 and 1 as input to the network. All augmentations are applied at a rate such that roughly 50% of images seen by each CNN during training contain artefacts. The order in which k-space augmentations are applied is important to best reflect the MR imaging process, with for instance RF spike → noise → lowpass filter/k-space sampling. In addition to k-space augmentation, image rotation, scaling and flipping augmentations are applied, as well as bias field augmentation to account for variation in image intensity across samples.

**RF spike artefact** is characterised by dark stripes over the image, as shown in Fig. 2 a) caused by the convolution of spikes in k-space of very high/low intensity during the FFT (Zhuo and Gullapalli, 2006). For augmentation we sample uniformly its location in k-space which specifies the angle/frequency of stripes and its magnitude which defines the intensity.

**k-Space noise** augmentation involves injecting Gaussian noise into the k-space, as shown in Fig. 2 b), to model Rician noise in the image domain. The desired signal-to-noise ratio (SNR) of the image is uniformly sampled between [-10dB, 30dB] and the corresponding amount of complex noise with zero mean and equal variance is added to the k-space.

**Blurring artefact** can be observed when acquiring data at lower resolution along one axis prior to resampling. Low-pass filter applied by truncating the k-space along one randomly chosen axis as shown in Fig. 2 c) can simulate this effect. The width of the filter defines the equivalent downsampling ratio, which is uniformly sampled between $2\times$ and $12\times$.

**Aliasing/wrap artefact** occurs when the imaging field of view (FOV) is smaller than the anatomy being imaged. This is retrospectively simulated by masking out k-space lines as shown in Fig. 2 d). A proportion of k-space lines are either randomly masked uniformly, or at regularly spaced intervals, along a random axis that defines the wraparound direction.

## 5. Experiments

**Proposed Network Architecture** All CNNs use the updated U-Net architecture from (Isensee et al., 2019) as their base architecture implemented in NiftyNet (Gibson et al., 2017). Each network is modified with two output heads, one for the segmentation $\hat{\mathbf{y}}$ and one for the uncertainty $\sigma^2$, where the uncertainty output from each CNN has a different number of channels – one for each decoupled uncertainty prediction. We first train the task network to learn a single task uncertainty $\sigma_t^2$. We then train $N$ teacher networks for each augmentation $i$, where each CNN outputs two uncertainties $\sigma_t^2$ and $\sigma_i^2$. Finally a combined network is trained to learn the task uncertainty and $N$ augmentation uncertainties. Each CNN is trained for 30,000 iterations with a patch size of $96^3$ and batch size of 2 across 4 GPUs with Adam optimiser (Kingma and Ba, 2015) and an initial learning rate of $10^{-4}$.

**Implementation Details** As in (Kendall et al., 2017), for numerical stability, each CNN is trained to predict log variance $s := \log \sigma^2$ instead of variance $\sigma^2$. In addition, the exponential mapping $\sigma^2 = \exp(s)$ enforces valid positive uncertainty values. We add a small constant $\epsilon$ to the variance to ensure the proper definition of the weighted cross entropy loss, $\mathcal{L}_{NN} = \text{CE}/(\sigma^2 + \epsilon) + \frac{1}{2}\log(\sigma^2 + \epsilon)$. It's value controls the network's sensitivity to noise and the amount of output uncertainty. For instance, if $\epsilon$ is small, the network is penalised more for making mistakes, outputting higher uncertainty to compensate. If $\epsilon$ is large, the amount the network is penalised is limited by $\text{CE}/\epsilon$ as $\sigma^2 \to 0$. Furthermore, smaller values of $\epsilon$ lead to training instability. Initially, $\epsilon$ is set to 0.05 and divided by 2 every time the loss

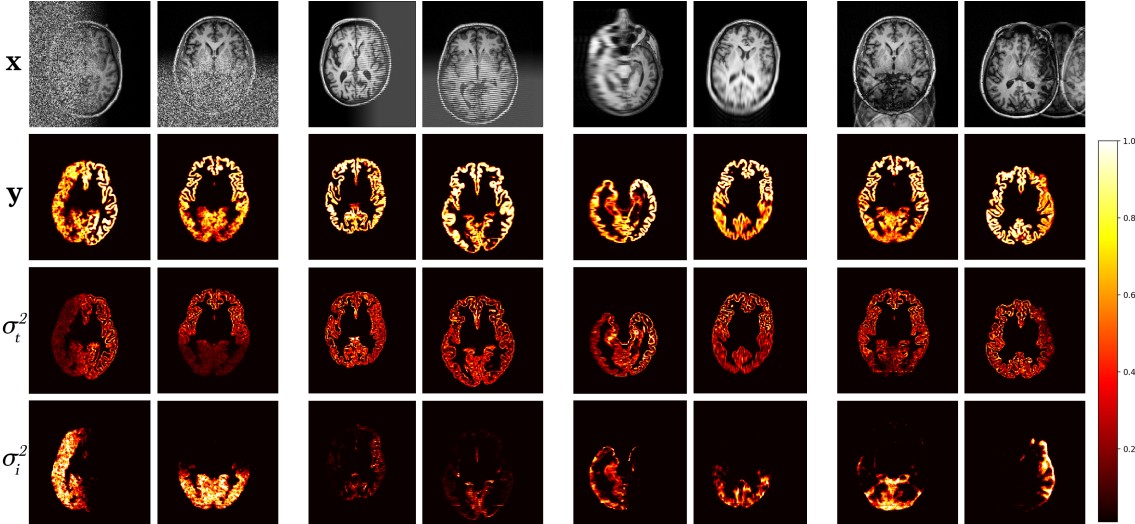

Figure 3: Qualitative results on a hold-out test set of simulated artefacts. First row: artefact-corrupted input image $\mathbf{x}$, second: resulting segmentation $\mathbf{y}$, third: predictive task variance $\sigma_t^2$, and fourth: corresponding augmentation uncertainties $\sigma_i^2 = \{\sigma_{noise}^2, \sigma_{rfspike}^2, \sigma_{blur}^2, \sigma_{wrap}^2\}$. Best viewed zoomed-in on digital copy.

plateaus until $\epsilon < 10^{-3}$. In parallel, at each of these steps, the learning rate is also halved.

**Training Data** for this work was obtained from the Alzheimer's Disease Neuroimaging Initiative (ADNI) (`adni.loni.usc.edu`). Launched in 2003, ADNI attempts to assess whether medical imaging and biological markers and clinical assessment can be combined to measure progression of Alzheimer's Disease. For training we use 272 MPRAGE scans that were deemed to be artefact-free, split into 80% train, 10% valid and 10% test. We evaluate our model on simulated and real-world artefacts in the task of grey matter segmentation.

**Simulated Data** A model trained with artefact augmentation was used to perform inference on the hold-out test set. Fig. 3 presents a selection of these results. For each sample, predicted segmentation, task and corresponding augmentation uncertainties are displayed. Areas of high uncertainty are generally in the artefacted regions. This enables us to quickly locate in the volume the image quality issue and judge its effect on the prediction by the level of uncertainty. Note, that in cases of heavy noise, the task uncertainty decreases as the signal is impaired, and therefore the model reverts back to the prior distribution.

**Real-world Data** Using the model trained on synthetic artefacts, we performed inference on a dataset of real-world artefacts identified as low quality by expert raters. A selection of these are shown in Fig. 4. We note that the uncertainty predictions generalise well to real-world artefacts and that the uncertainty is generally higher in the artefacted regions.

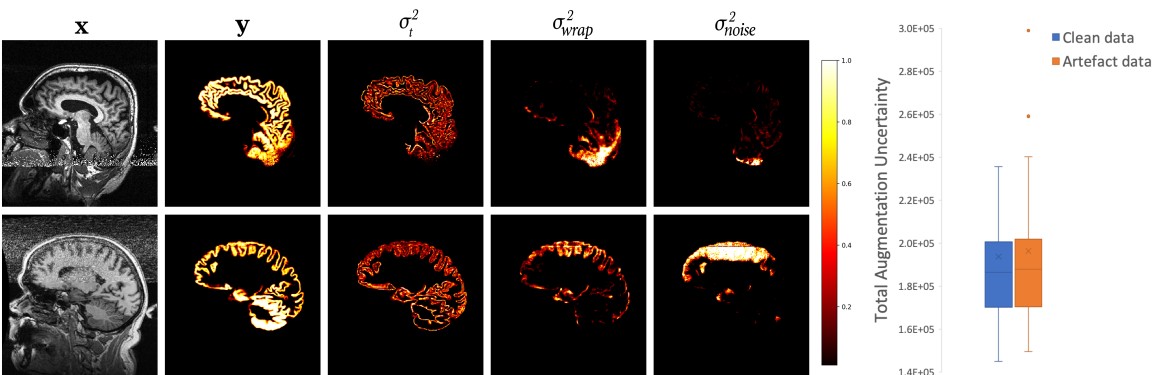

Figure 4: Left: Real-world artefact results. From left to right: artefact-corrupted input image $\mathbf{x}$, predicted segmentation $\mathbf{y}$, task uncertainty $\sigma_t^2$, and corresponding augmentation uncertainties $\sigma_{wrap}^2$ and $\sigma_{noise}^2$. Right: Total augmentation uncertainty over the data. View zoomed-in on digital copy.

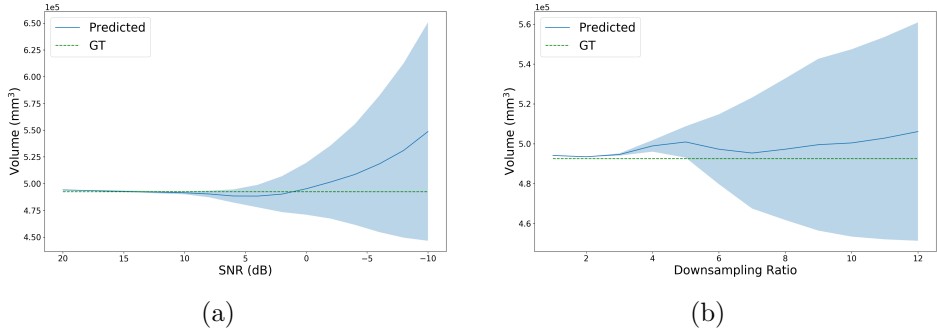

Figure 5: Predicted confidence intervals at $\pm\sigma$ on volume measurements from the grey matter segmentation for images with a) increasing noise and b) increasing blur.

**Entropic Uncertainty** The per-voxel variance values predicted by the network pertain to the logit space. While these values directly are useful indicators of uncertainty given the data, they relate to the actual uncertainty in the segmentation prediction via the entropy. Therefore, we can estimate a measure of uncertainty in our probabilities by computing the entropy as $\mathcal{H} = -\sum_c p_c \log p_c$, where $p_c = p(\mathbf{y} = c|\mathbf{f}^{\mathbf{W}}(\mathbf{x}), \sigma)$ are the scaled output logits from the network. Furthermore, using standard variance-entropy relations (Jee and Ratnaparkhi, 1986) we can obtain approximate error bars on segmentation measurements. In Fig. 5, we use our uncertainty predictions to estimate the confidence interval for increasing noise and blurring in the image, relative to clean data. As this is a relative measure of uncertainty, i.e. even noise-free images will have some level of predicted uncertainty, we must transform the uncertainty by some amount to obtain calibrated error bars. In this work we simply compute the difference from the known uncertainty of a noise-free image, but these scaling parameters could be learnt from the data as in (Eaton-Rosen et al., 2019).

## 6. Discussion

The aim of this work was to build a deep-learning framework capable of identifying and decoupling sources of uncertainty due to MRI artefacts that may affect a given segmentation task. We have shown that it is possible to obtain approximately decoupled uncertainties that reflect the presence (location and severity) of k-space artefacts. We have also shown that these uncertainties can be used to generate error bars on segmentation measurements. **Limitations** Our uncertainty predictions are limited by the fact that we are estimating uncertainty given the data $p(\mathbf{y}|\mathbf{x}, \sigma)$, but have not modelled the likelihood of the data itself $p(\mathbf{x})$, achievable through methods such as autoencoders. Extrapolation problems also limit our ability to decouple uncertainty, as the introduction of extreme artefacts results in an unstable learning process. Lastly, we assume that the original data (used for training the task network) is artefact-free, possibly resulting in inflated task uncertainty estimates.

## 7. Conclusion

We have presented a method for estimating quality-induced task-specific uncertainty using a heteroscedastic noise model. Entirely self-supervised, the proposed model can approximately decouple and localise sources of uncertainty related to different MRI artefacts, thus automatically highlighting problematic areas affecting segmentation predictions. The method is general and may be applied to other automated image analysis processing tasks.

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
