# OpenReview forum: "A Heteroscedastic Uncertainty Model for Decoupling Sources of MRI Image Quality"
_MIDL.io/2020/Conference — MIDL 2020_

### Official Review · AnonReviewer4 · 2020-03-02
**interesting ideas for modeling image quality**

**Rating:** 3
**Confidence:** 4
**Recommendation:** Poster

**Summary:**

This paper outlines a multi-stage student-teacher CNN model for decoupling different sources of image quality issues in MRI data. This is a important topic and as the authors correctly note, image quality is task dependent, therefore modelling uncertainty for a tissue-class segmentation task makes sense for neuroimaging applications.

**Strengths:**

- the evaluation using different types of localized artifacts in figure 3 is convincing.
- performing the artifact simulation in k-space makes sense.
- the student-teacher network approach is novel, although potentially would benefit from a comparison with a more straight forward approach.
- the pixel-wise uncertainty measure may be useful for some tasks.

**Weaknesses:**

This paper is a little light on validation on 'real-world' artifacts e.g. the quantitation in figure 4 doesn't distinguish between different types of artifacts, which was the main justification for the model design.

It is also not clear what the simulation in figure 5 is attempting to illustrate. It would be useful to have more detail on how this was carried out and why it is important included in the methods or the results section.

**Justification Of Rating:**

Using aleatoric uncertainty on a segmentation task for image quality control is a novel idea and potentially useful for neuroimaging research studies where segmentation is often a crucial post processing step. The framework outlined here could be expanded to other tasks. One downside to this paper is an absence of a strong validation on real-world artifacts.

**Paper Type:**

methodological development

**Special Issue:**

no

---

> ### Author Response · Authors · 2020-03-27
> **Response to AnonReviewer4**
>
> We would like to thank the reviewer for their comments. Regarding the weaknesses identified:
>
> 1) This paper is a little light on validation on 'real-world' artifacts e.g. the quantitation in figure 4 doesn't distinguish between different types of artifacts, which was the main justification for the model design.
>
> We agree with the reviewer that more validation could have been done on real-world artefacts, but unfortunately at the time of writing we were limited by the data we had available. We simply did not have enough data with the specific artefact subtypes that we trained on. We are currently in the process of gathering more real-world artefact data with QC rater’s reports/comments, allowing us to test on specific artefact subtypes for further validation in a journal paper version of this work.
>
> Regarding fig 4, indeed we only distinguish between artefact data and clean data, but this shows that our model roughly decouples task and artefact uncertainty, which is also one of the main justifications of our model design. However, we must note that because ADNI QC raters label images as artefacted if they have an artefact anywhere in the image, the difference in total artefact uncertainty on clean and artefact data is not that significant. Thus, if the artefact is not in the brain (e.g. neck region), the CGM segmentation uncertainty will be low, but the image is labeled as artefacted. This means that the segmentation uncertainty can be similar between “clean” and “artefacted” data as can be seen in fig 4. Most ADNI images with quality issues have this kind of behaviour, so the plot on fig 4 has a complex interpretation.
>
> 2) It is also not clear what the simulation in figure 5 is attempting to illustrate. It would be useful to have more detail on how this was carried out and why it is important included in the methods or the results section.
>
> We agree with the reviewer that more detail could have been included in this section but we had limited space in the MIDL paper. Fig 5 is attempting to validate our uncertainty predictions by exploring how the predictive uncertainty changes as the input image degrades in quality. To do this we increased the amount of simulated noise/blurriness gradually and performed inference on each image sample using our trained model. Fig 5 illustrates that the predicted uncertainty goes up with decreasing SNR (increasing noise) and downsampling ratio (increasing blurriness), therefore demonstrating that the model’s predictions reflect the quality of the input image. It also demonstrates that we can use the predicted uncertainty to obtain error bounds on segmentation measurements, as this is likely to be more useful from a clinical point of view than the raw voxel-wise uncertainty values, and that these error bounds are reasonable and incorporate the ground-truth volume within the error bar itself. This means that the uncertainty predictions are reasonably well calibrated and the model understands when it cannot accurately predict the segmentation given the quality of the image. We will further clarify this in the final version of the paper.

---

### Official Review · AnonReviewer3 · 2020-03-07
**A theoretically grounded paper on QC for MR images**

**Rating:** 4
**Confidence:** 3
**Recommendation:** Oral

**Summary:**

The paper proposes a solution to Quality Control (QC) in MR images with the artifact by modelling the problem as a heteroscedastic uncertainty estimation for different artifacts individually. The proposed method is theoretically grounded, limitations of the work and assumption are clearly stated.  Experiments on a simulated and a real-world dataset show the usefulness of the method.

**Strengths:**

+ The problem and the proposed method are well motivated.
+ Decoupling of multiple uncertainties seems like a clever idea.
+ Results show that when the network is trained on the simulated data it is able to decouple artifacts in real-world data.
+ Good qualitative results.


**Weaknesses:**

- A combination of different loss-term (L1, L1 on the gradient, and SSIM) is used for consistency between teacher-student network uncertainties. It would be nice if the effect of each of this loss-term was evaluated separately.
- As the predicted uncertainty is heteroscedastic (i.e. input dependent and not task-dependent), it would be nice to see if the learnt uncertainty generalizes to other segmentation tasks without using K-space artifact augmentation during training.
- Though qualitative images are good and show the usefulness of the method, it would be good if better quantitative results for each task uncertainty and how it can be used for automatic QC was provided.

**Justification Of Rating:**

The paper proposes a novel method for artifact estimation individually for different sources of artifacts. Results are promising and show the effectiveness of the method. Experiment on the real-world data shows the applicability of the method.

**Paper Type:**

methodological development

**Special Issue:**

yes

---

> ### Author Response · Authors · 2020-03-27
> **Response to AnonReviewer3**
>
> We would like to thank the reviewer for their comments. In response to the reviewer's concerns:
>
> - A combination of different loss-term (L1, L1 on the gradient, and SSIM) is used for consistency between teacher-student network uncertainties. It would be nice if the effect of each of this loss-term was evaluated separately.
>
> As stated in our response to point 4 from AnonReviewer2, the choice of consistency losses was empirically driven and based on a common choice of loss functions for pixel-wise regression problems, and the fact that we found an L2 consistency loss inadequate.  Indeed, for completeness, this would be an interesting set of experiments to do, but requires us to retrain the model with the different loss functions separately. We will, however, do this for the journal paper based on this work.
>
> - As the predicted uncertainty is heteroscedastic (i.e. input dependent and not task-dependent), it would be nice to see if the learnt uncertainty generalizes to other segmentation tasks without using K-space artifact augmentation during training.
>
> As stated in the methodological introduction of the paper, the predicted uncertainty is actually task-specific, so it would need to be retrained for every new task, i.e. the uncertainty is dependent on the quality of the input data, but is still specific to the task that we trained for - CGM segmentation on ADNI data. It is likely to generalise to other datasets, but it should not generalise to another task by definition.
>
> - Though qualitative images are good and show the usefulness of the method, it would be good if better quantitative results for each task uncertainty and how it can be used for automatic QC was provided.
>
> - Under the assumption that the user want to know a task-specific QC metric, the sum of the uncertainty over all pixels would be the relevant metric to use here. Note however that this metric will be related to the network's ability to solve for this task and not the visual quality of the image itself. For the latter, we are working on a separate paper that is currently under review at MICCAI.

---

### Official Review · AnonReviewer1 · 2020-03-13
**Nice work. Despite the lack of validation the novelty is undeniable**

**Rating:** 4
**Confidence:** 4
**Recommendation:** Oral

**Summary:**

The authors present good work that aims to automate quality control in MRI images. As a quality measure, the authors use the voxel uncertainty associated with a segmentation task, in this case, grey matter segmentation. The uncertainty of this task can have different sources, and this fact is the main novelty of the article since the authors estimate the uncertainty due to different artefacts.

**Strengths:**

- The employment of cascading teacher-student networks, which allow creating "surrogate truth" of the uncertainties per image artifact which reinforce the uncertainty estimation
- The adaptation of the framework described by Kendall and Gal to estimate the uncertainties from a multi-task perspective without the employment on any kind of uncertainty label to the aforementioned architecture
- The adoption of the cascading teacher-student networks plus the uncertainty framework which allow strongest model regularization due to the  employment of uncertainties form the augmentation process as uncertainty labels


**Weaknesses:**

- Lack of validation
- The approach is just tested for one dataset
- The assumption of independence of the difference variances
- Facilitate understanding of the work
- The quantitative results are clearly insufficient and unclear
- The description for obtaining the entries is missing

**Justification Of Rating:**

In the field of medical image processing, it is common to find small adaptations of methods coming from the field of computer vision, however, this work goes further and proposes novel approaches to these methods that are also of clear clinical interest.

**Paper Type:**

methodological development

**Questions To Address In The Rebuttal:**

- The similarities of the method proposed with a multitask approach like the one proposed by Kendall [] are evident, equation 5 is an example of them. For the sake of clarity, could you list the differences?

- Each augmentation network provides an uncertainty which is employed later as "label", do the results get much worse if we only use these and omit "NetCombined"?  Could you show some comparisons?

- Could you provide a table of results by combining the different factors of the additional loss term (including $\lamda$)?

- A big limitation has not been included. Generalization, could you provide results employing a different dataset(s)? Please mention how you could handle Domain Adaptation employing your method.

- Could you properly explain the right part in figure 4? How is it possible that clean and augmented data have a similar uncertainty?

-  It would be advisable to use a paragraph to briefly introduce student-teacher networks since are not common in the medical imaging field.

- The work is nice but figure 1 does not properly illustrate the method, could you please modify it?

- Figures 3 and 4 say "corresponding augmentation uncertainty $\sigma^2_i$ ". Please, enumerate them.


[1]  Kendall, A., Gal, Y. & Cipolla, R. Multi-Task Learning Using Uncertainty to Weigh Losses for Scene Geometry and Semantics. in Proceedings of the IEEE Conference on Computer Vision and Pattern Recognition (CVPR) 1–10 (2018).

**Special Issue:**

yes

---

> ### Author Response · Authors · 2020-03-27
> **Response to AnonReviewer1**
>
> We would like to thank the reviewer for the comments and for recognising the novelty of the proposed method and the potential clinical interest. A concern raised by the reviewer is that there is a lack of validation in the paper. Indeed we only presented results for CGM segmentation, but we also trained and tested on synthetic data (simple random shapes with added artefacts) when prototyping the method, however, we didn’t include these results in the paper due to limited space. We also feel that the proposed method is quite general and there is no reason it couldn’t be applied to other datasets or even non-medical problems. We hope to include further validation on other datasets in a future journal publication of this work. We would like to stress that the assumption of conditional independence is an assumption of our model.
>
> Regarding the reviewer's questions:
>
> 1) The similarities of the method proposed with a multi-task approach like the one proposed by Kendall are evident, equation 5 is an example of them. For the sake of clarity, could you list the differences?
>
> - The teacher-student learning process
> - The proposed loss function with the added artefact constraint
> - The MRI artefact simulation ecosystem
> - All the results when applied to medical data
> - Demonstration that there is transferability of features from synthetic to real data
>
> 2) Each augmentation network provides an uncertainty which is employed later as "label", do the results get much worse if we only use these and omit "NetCombined"?  Could you show some comparisons?
>
> This is an interesting point. Actually, a “noise-only teacher” artefact network predicts noise uncertainty better than “netCombined”. This is because the teacher network only has to learn the appearance of noise artefacts, whereas “netCombined” must learn all possible artefacts, which, with limited network capacity, tends to over-smooth the output space of predicted uncertainty. However, the “noise-only teacher” fails to accurately predict any uncertainty if presented with images with two types of artefacts, while “netCombined” does not. Thus, the purpose of “netCombined” is to provide predictions of uncertainty when one or more artefacts are present, something which the teacher networks cannot do.
>
> 3) Could you provide a table of results by combining the different factors of the additional loss term (including)?
>
> As per the previous question, a network trained with only noise artefacts will fail if presented with an image with blurry artefacts. The dice scores are very poor because the network has never seen such data, which is expected. We thus decided to omit the results from the paper. Nonetheless, we will add a sentence on the final version of the paper to explain this specific point.
>
> 4) A big limitation has not been included. Generalization, could you provide results employing a different dataset(s)? Please mention how you could handle Domain Adaptation employing your method.
>
> This is a really interesting point. The proposed method was indeed developed using a single dataset. We have shown transferability of features from synthetic to real data, but have not shown that it’s domain transferable. Nonetheless, the proposed network can be extended using many of the Domain Adaptation techniques (e.g. adversarial, consistency-based, etc), so we do not foresee an issue here. It is an aspect we would like to explore in future work.
>
> 5) Could you properly explain the right part in figure 4? How is it possible that clean and augmented data have a similar uncertainty?
>
> Actually, this is an issue with the labeled data itself. ADNI QC raters label images as artefacted if they have an artefact anywhere in the image. Thus, if the artefact is not in the brain (e.g. neck region), the CGM segmentation uncertainty will be low, but the image is labeled as artefacted. This means that the segmentation uncertainty can be similar between “clean” and “artefacted” data. Most ADNI images with quality issues have this kind of behaviour, so the plot on fig 4 has a complex interpretation. We are currently tackling this issue by modelling quality as an artefact removal regression task, where preliminary results can be seen in the following linked image (https://tinyurl.com/yxy38zyu). These results are part of a different paper currently under review.
>
> 6) It would be advisable to use a paragraph to briefly introduce student-teacher networks since are not common in the medical imaging field.
>
> Indeed, and we thank the reviewer for pointing out this omission. We will do so in the final version.
>
> 7) The work is nice but figure 1 does not properly illustrate the method, could you please modify it?
>
> We will try to add more detail to the figure, but the network and training methodology is quite complex to represent pictorially.
>
> 8) In figures 3 and 4 we will enumerate the corresponding augmentation uncertainties.

---

### Official Review · AnonReviewer2 · 2020-03-17
**Review of "A Heteroscedastic Uncertainty Model for Decoupling Sources of MRI Image Quality"**

**Rating:** 2
**Confidence:** 4

**Summary:**

This paper seeks to decouple some of the possible sources of noise in MRI acquisition, via a model that treats the noise sources as independent and thus the associated variances as additive. Using this idea and simulated artefacts, the authors present proof of concept results related to the task of image segmentation. The results are of a qualitative nature.

**Strengths:**

The main theme of this paper, that of training neural networks to provide measures of uncertainty associated with candidate image artefacts is innovative. The authors carry this out via simulated "augmentations" in k-space (fake image artefacts) that are used to sequentially train 3 different networks (Fig. 1). Should the assumption of independence of the effect of the artefacts on MRI image intensity hold, the strategy to design appropriate loss functions seems appropriate. The idea of applying the simulated artefacts in k-space and then inverse FFT'ing to get simulated MRI artefacts is reasonable.

**Weaknesses:**

I found this to be a bit of a "seat of the pants" approach, where the assumption of independence is not clearly justified in any way, and yet, is the entire premise of the method. The authors state "While interactions with task uncertainty (task harder to learn with noisier data) or between degradation types (blurring and noise for instance) exist, their modelling would require the learning of new covariance terms and would greatly complexify both model and training procedure." I think they hit upon the key issues here. How would the effect of image artefacts truly be independent? Surely in even the most simple scenarios, the effects of blurring, RF noise etc. could co-occur. I also found there to be no attempt to present the experimental results with quantitative measures or analyses. The qualitative examples here do not convince me that the predicted task uncertainties, with respect to grey matter segementation, are correct, or that a much simpler image processing method would not do the trick.

**Detailed Comments:**

In addition to the questions raised above, which if you could address would help strengthen the paper, here are some further suggestions:

1) In general, the main premise of this paper is clear and the ideas are well motivated, but there are a lot of details in the formulation that are not clearly justified. Better motivating the choices for loss function construction and relative weighting would help. Ablation studies to show stability with variations in loss function weightings or details, would help too, as is by now standard for vision, medical imaging and machine learning papers.

2) Can you think more carefully about how you would validate this approach and not just present estimates of variances and confidence intervals? Ultimately your goal, as I understand it, is to help in QC for MRI assessment for large scale studies, based on training on a sufficient number of degradation cases. In the present article the results are at best proof of concept.

**Justification Of Rating:**

I think the authors are on to an interesting problem, and the basic problem this paper seeks to solve is both valid and important. However, the assumptions made are not clearly justified though, and aspects of the design are not easy to assess. In essence, the authors assume that the effects of artefacts are by nature decoupled, when the title of the paper suggests that the article will provide a means for doing this. In the formulation (Fig. 1 and associated text) I do understand that 3 separate networks were trained, with the losses coupled and with additional terms introduced to penalize differences in uncertainty predictions, etc. But rather than just saying how these loss functions are heuristically designed it would help to better motivate the details and the design choices.


**Paper Type:**

both

**Questions To Address In The Rebuttal:**

1) In the introduction you discuss voxel-wise segmentation as the task being investigated here, but in the actual experiments you only treat grey matter segmentation. Why can't you evaluate under a more challenging task, e.g., something more complex than binary segmentation?
2) Can you better motivate the premise, i.e., give clear examples to show that at least some of the various MRI degradation possibilities you consider lead to "independent" artefacts?
3) What happens when more than one artefact is present at the same time? Does performance degrade gracefully? Are aspects of the formulation still valid?
4) I didn't fully understand the manner in which the additional loss terms are added to construct the loss term for augmentation (eqs 6 and 7). There seems to be some engineering going on here. It is not clear how the cross-entropy loss is weighted with these new sequentially added terms. Can these ideas be put on more solid theoretical footing?
5) Can you, in your experiments, provide quantitative analyses to show validity of the predictions? I found it difficult to assess the quality of your predicted variances, and the correctness of your confidence intervals. I think you really need to think a lot more carefully about validation.

**Special Issue:**

no

---

> ### Author Response · Authors · 2020-03-27
> **Response to AnonReviewer2**
>
> We would like to thank the reviewer for their helpful comments. Addressing the reviewer’s questions:
>
> 1) We chose the task of grey matter segmentation purely due to simplicity of the visualisation, quantification and model training. It is also a dataset for which we have high-quality QC’d images (our method assumes that the data must be free from artefacts) and corresponding segmentation labels. But we are interested in evaluating the method on more challenging problems in the future.
>
> Note that the model is scalable to any segmentation task with arbitrary label complexity as it optimises for cross-entropy. This means the method should apply, not only to binary classification problems, but also to multi-class segmentation, where the predicted variance reflects the uncertainty of a predicted class. Note that all our assumptions of uncertainty decomposition made in the paper still hold in this case.
>
> 2) We agree with the reviewer that we could have provided better motivation for the paper, specifically regarding the MRI degradation possibilities. In terms of MRI acquisition, as opposed to their appearance in the final image, patient motion and coil noise are clearly independent effects that cause artefacts. Furthermore, from a physics acquisition point of view for example, the point spread function (PSF) is dependent on the image resolution and window size, while noise is a product of the receiver coils, meaning that blurriness due to low resolution is independent of the coil noise and sensitivity. At the link (https://tinyurl.com/tzdz6ll) we show an image with a large PSF, the same image with low PSF but with noise, and the same image with a large PSF and noise. Note that the last image shows both effects.
> However, note that we have now revised the paper to state that “we assume that the artefact processes are conditionally independent given the artefact label”, as we believe this is a more precise description of the setup.
>
> 3) This is an interesting and complex question. The network’s ability to disentangle multiple artefacts is not only dependent on network capacity and the kinds of artefacts seen during training, but is also conditioned on the presence and severity of each artefact. For example, even though the two physics processes are independent, the network’s ability to estimate the noise level is hindered by the presence of extreme blurriness. Note, however, that the network will still detect the presence of artefacts, but likely not to the degree that they are present in the image due to how they interact. For instance, the network may output uncertainty for both noise and blurriness, just at a lower value due to their interaction. Diagonal covariance terms could be included to capture this interaction, but this is likely difficult for the network to learn from the magnitude image information alone. However, this is indeed an important point, thus, we will highlight this limitation in the discussion of the paper.
>
> 4) Indeed, the choice of loss functions was empirically driven. The loss of the segmentation is a cross-entropy, while the supervising loss for each artefact network is defined as an L1 + Structural Similarity (SS) + L1(image gradients). We initially tried L2, but found this produced over-smoothed uncertainty estimates, while including L1’s and SS produced sharper uncertainty predictions. This is a common choice of loss terms for pixel-wise regressions problems (for example in “Loss Functions for Image Restoration with Neural Networks.” Zhao et al. 2018). The terms are added to each other with no weighting. In theory, they could be weighted to further improve performance, but the computational complexity of that hyperparameter tuning made this experiment hard to perform. Note, however, that the fact that the model trained and produced good results with no weighting nor parameter tuning means that the model is robust and stable w.r.t these parameters. Also note that the predicted uncertainty might need to be re-scaled (e.g. “As easy as 1, 2... 4?” Eaton-Rosen et al. MICCAI 2019) to appropriately capture the full uncertainty, as most estimated uncertainties using deep learning are uncalibrated.
>
> 5) Indeed, due to page limitations, we have omitted some experiments. In the current work, the only validation of the uncertainty quality itself is in fig 5, where we show that the uncertainty estimates do indeed capture the ground-truth segmentation for the full range of SNR and blurriness. We are currently working on a journal submission of this work where we seek to demonstrate the quality of the uncertainty estimates. We demonstrate that when only one type of uncertainty goes up, the other uncertainties are largely unaffected (https://tinyurl.com/rfloxat). We also show that estimated uncertainty does indeed capture the truth with the right statistics, but these experiments were not in the MIDL submission due to the soft page limit.

---

### Meta-Review · Area_Chair1 · 2020-04-06
**MetaReview of Paper171 by AreaChair1**

**Rating:** 4
**Recommendation For Accepted Papers:** Oral, Poster

**Metareview:**

This paper tries to decouple some sources of MRI artifacts to assess image quality.

All reviewers agree in the novelty of the paper and the results are sound although somehow limited due space limitation.

Interesting paper worth to be presented in the MIDL

**Paper Type:**

methodological development

**Special Issue:**

yes

---

### Decision · Program_Chairs · 2020-04-11

Accept